# The Ultra-Processed Food Content of School Meals and Packed Lunches in the United Kingdom

**DOI:** 10.3390/nu14142961

**Published:** 2022-07-20

**Authors:** Jennie C. Parnham, Kiara Chang, Fernanda Rauber, Renata B. Levy, Christopher Millett, Anthony A. Laverty, Stephanie von Hinke, Eszter P. Vamos

**Affiliations:** 1Public Health Policy Evaluation Unit, School of Public Health, Imperial College London, London W6 8RP, UK; chu-mei.chang@imperial.ac.uk (K.C.); c.millett@imperial.ac.uk (C.M.); a.laverty@imperial.ac.uk (A.A.L.); e.vamos@imperial.ac.uk (E.P.V.); 2Center for Epidemiological Research in Nutrition and Health, University of São Paulo, São Paulo 01246-904, Brazil; rauber@usp.br (F.R.); rlevy@usp.br (R.B.L.); 3Department of Preventive Medicine, School of Medicine, University of São Paulo, São Paulo 01246-903, Brazil; 4Public Health Research Centre & Comprehensive Health Research Center (CHRC), National School of Public Health, NOVA University of Lisbon, 1600-1500 Lisbon, Portugal; 5School of Economics, University of Bristol, Priory Road Complex, Bristol BS8 1TU, UK; s.vonhinke@bristol.ac.uk

**Keywords:** ultra-processed food, school lunch, child and adolescent nutrition

## Abstract

British children have the highest levels of ultra-processed food (UPF) consumption in Europe. Schools are posited as a positive setting for impacting dietary intake, but the level of UPFs consumed in schools is currently unknown. This study determined the UPF content of school food in the UK. We conducted a pooled cross-sectional analysis of primary (4–11 years, *n* = 1895) and secondary schoolchildren (11–18 years, *n* = 1408) from the UK’s National Diet and Nutrition Survey (2008–2017). Multivariable quantile regression models determined the association between meal-type (school meal or packed lunch) and lunchtime UPF intake (NOVA food classification system). We showed that on average, UPF intake was high in both primary (72.6% total lunch Kcal) and secondary schoolchildren (77.8% total lunch Kcal). Higher UPF intakes were observed in packed lunch consumers, secondary schoolchildren, and those in lower income households. This study highlights the need for a renewed focus on school food. Better guidance and policies that consider levels of industrial processing in food served in schools are needed to ensure the dual benefit of encouraging school meal uptake and equitably improving children’s diets.

## 1. Introduction

British children consume 65% of their daily calorie intake as ultra-processed foods (UPF) [1], the highest level in Europe [2]. UPFs are designed to be hyperpalatable, convenient, non-perishable, and relatively cheap [3]. As such, UPFs are often higher in fat, salt, and sugar and have an altered food structure which makes them more digestible, less satiating and have a higher glucose potential than less processed foods [4,5,6]. Research has highlighted a range of negative health impacts associated with UPFs. High UPF consumption in children is associated with poorer dietary quality [7,8], but negative effects of UPF consumption have also been shown to be possibly independent of diet quality [9,10]. In children, UPF is associated with an increased risk of being overweight or obese in later life [11,12]. In adults, high levels of UPF consumption are associated with obesity, type 2 diabetes [13], cancer [14], cardiovascular disease [15], and premature death [16].

Schools have been proposed as an important area for equitably improving children’s diets [17]. In the UK, school food must comply to mandatory food-based guidelines, and meals are given free only to low-income children and to children below the age of 7 years in England and 9 years in Scotland [18] We refer to food provided by the schools as ‘school meals’. However, the quality of school food in the UK is debated. On average, school meals have been shown to have a preferable nutritional profile compared to packed lunches (food brought from home) [19,20,21,22]. However, concerns have been raised about the quality of school meals, which although preferable to packed lunches, are not yet optimal [23,24]. Furthermore, there has been little research examining the extent to which school meals or packed lunches include UPFs. This is an important gap in the literature; as school meals are publicly funded, they should contain minimal quantities of food products known to be harmful to child health. It is imperative that we understand UPF consumption in the school setting, in order to better guide policy makers and parents in reducing UPF intake and to improve the quality of food consumed in schools.

This study aims to describe the UPF content of school food in the UK among primary and secondary schoolchildren between 2008 and 2017 and to explore differences between school meals and packed lunches. The study additionally aims to compare the UPF content of school food consumed by children with different household incomes.

## 2. Materials and Methods

### 2.1. Data Source

This study used data from the years 2008–2017 of the National Diet and Nutrition Survey (NDNS) [25], which is a rolling programme of cross-sectional surveys aiming to provide representative snapshots of dietary intake in the UK among those aged 1.5 years and over. The survey used a multi-stage random sampling method to select households from a list of postcodes; further details are published elsewhere [26]. Trained interviewers collected sociodemographic information through interviews and administered dietary diaries to participants. To account for weekly and seasonal variations, dietary diaries were administered so there was an even number of days and months across the sample. The participants were instructed to record the location, time and quantity of all food and drink consumed over three or four consecutive days. Diaries were filled out by a guardian for the participants aged 12 years or younger. The interviewer returned to collect the diary and reviewed the data with the participant. The years 2008–2017 were pooled in this study to maximise sample size.

### 2.2. Study Participants

All NDNS participants between the ages of 4–18 years attending a primary or secondary school were included in the initial sample selection (*n* = 4800). Of this initial sample, 1479 (31%) were excluded as they did not record a lunchtime intake whilst at school. Three participants were removed due to missing ethnicity data, and 16 were removed for missing meal type (school lunch or packed lunch) data, leaving a final analytical sample of 3303 participants. School lunches were defined as food items consumed Monday–Friday between 12:00 and 14:00 on school premises. The total number of school lunches recorded by participants varied from one to four days (1 day [*n =* 584], 2 days [*n =* 1379],3 days [*n =* 786], and 4 days [*n =* 554]). Not recording a school lunch was likely due to data collection during a school holiday; this was affirmed by a higher prevalence of non-recording in the months of August and December. However, for older children who were permitted to leave school premises, this may have indicated that they purchased and consumed lunch externally.

### 2.3. Exposure: School Lunch Meal Type

The dietary diaries recorded the meal type for each food item; that is, whether it was consumed as part of a school meal or packed lunch. If the item was described as ‘food from home’ it was categorised as a ‘packed lunch’ and if it was described as ‘bought at the canteen’ in the dataset it was categorised as a ‘school meal’. If the meal type of a school lunch was not recorded consistently for every food item (*n* = 1580 participants), a survey question asked in the interview ’on a school/college day, what do you/does (child’s name) usually have for lunch?’ was used to determine the child’s meal type. Participants were classified as either school-meal or packed-lunch consumers accordingly. For participants who had complete records of meal type and survey response (*n* = 1554), there was a high level of agreement between the two measurements (91%).

### 2.4. Outcome: Ultra-Processed Food

The degree of food processing in items consumed during school lunches was described using the NOVA food classification system, which includes four main food groups [7,27], as described in Table 1

The main outcome variable was UPF consumption (NOVA 4). The weight (grams) and energy content (kcal) from each NOVA category consumed at lunch was calculated and averaged per school lunch for each participant. The level of UPF consumed at lunch was expressed relative to the total weight (% of total grams (% g)) and energy (% of total Kcal (% kcal)) consumed at lunch, consistent with previous research in this field [28]. It was important to use both contribution to weight and energy intake to avoid masking any differences in energy density across food and drink categories.

Additionally, the consumption of subsidiary NOVA categories was analysed. NOVA 1 subgroups (minimally processed drinks, fruit and vegetables, dairy products, starchy products, minimally processed meat and fish products) and the NOVA 4 subgroups (ultra-processed drinks, ultra-processed bread, snacks, condiments, puddings and desserts, fast foods [pizza, burgers, chips], ready-to-eat dishes, yogurt and milk drinks, cheese, meat and fish, processed vegetables [baked beans]) are presented in Table 2. The NOVA 1 and NOVA 4 subgroups accounted for the majority of dietary intake in school lunches (>94%). The NOVA 2 (processed culinary ingredients) and NOVA 3 (processed foods) subgroups are not presented due to their low average consumption (mean lunchtime intake 1.3% kcal and 7.1% kcal, respectively).

### 2.5. Covariates

Study covariates included were survey year (2008–2017), sex (male/female), age (4–18 years), ethnicity (white/ethnic minorities), equivalised household income (low/middle/high), quintile of Index of Multiple Deprivation (IMD), region (North England, Central/Midlands, South England (incl. London), Scotland, Wales, Northern Ireland). IMD is an area-level measure of deprivation. Equivalised household income was imputed for participants with missing data (*n* = 137) using ten iterations of the classification and regression trees (CART) method [29] in R.

### 2.6. Statistical Analysis

Sample characteristics were compared across meal type and school phase (primary [ages 4–11 years] and secondary [ages 12–18 years]) using *t*-tests and χ^2^ tests, as appropriate. Due to the skewed distribution of UPF variables, a Kruskal–Wallis test was used to examine the unadjusted difference in UPF consumption across population subgroups, stratified by school phase. The median intake of UPFs was further computed for each combination of meal type and household income group, and presented graphically. In addition, the average contribution of each MPF and UPF sub-category to children’s overall lunchtime intake is presented graphically by meal type and school phase for comparison.

Quantile regression was used to explore the difference in UPF content between school meals and packed lunches, stratified by school phase. Quantile regression was used, as the assumption of normally distributed residuals required for linear regression was violated. As quantile regression estimates the median (or other quantile) of the outcome distribution instead of the mean, it is less sensitive to the influence of outliers. Covariates were included in the model in two stages: Model 1 included age and sex and Model 2 additionally included survey year, ethnicity, region, IMD, and income. Marginal effects from a quantile regression model that included an interaction term between meal-type and income are presented graphically.

Individual consumption of each MPF and UPF sub-categories were dichotomised into consuming none (0 g/lunch) or some (>0 g/lunch) and logistic regression was performed to compare the likelihood of consuming each sub-category of MPF/UPF between meal type and stratified by school phase. Results presented were fully adjusted for all covariates.

Sensitivity analyses were performed to check for the robustness of findings. Firstly, additional adjustments for total energy intake, total grams intake and BMI were performed, to check for robustness against confounding influence of variation in body size and energy intakes. Secondly, 1580 participants whose meal type was not consistently recorded in the dietary diary were excluded. Thirdly, the impact of dietary misreporting was estimated using the Goldberg method, adapted for children [30,31]. Unreliable energy reporters (*n* = 485, 15%) were identified by comparing a participant’s estimated energy requirements (Schofield equations) to their reported energy intake and excluded.

All statistical analyses were performed using R (version 4.0.2). Survey weights were applied in all data analyses to account for sampling and non-response bias [32]. *p*-values of <0.05 were considered statistically significant for all tests.

## 3. Results

In the sample of 3303 participants, 57% were in primary school and 47% ate school meals, while 53% ate a packed lunch (Table 3). Overall, children who had a school meal were more likely to be younger, be from an ethnic minority, have a lower household income, and be from a more deprived neighbourhood than children who took a packed lunch. There was a significantly higher median intake of UPF (% kcal) at lunch in secondary schoolchildren than primary schoolchildren (77.8% kcal vs. 72.6% kcal) (Appendix A). Additionally, the median intake of UPF (% kcal) was consistently lower in school meals than packed lunches across primary (61.0% kcal vs. 81.2% kcal, respectively) and secondary schoolchildren (70.1% kcal vs. 83.5% kcal). These patterns were similar when the contribution of UPF towards total weight intake (UPF % g) was analysed. The median UPF consumption (% g) was lower in primary schoolchildren than secondary (37.7% g vs. 59.7% g) and lower in school meals than packed lunches in both primary and secondary schoolchildren (Appendix A). Furthermore, these relationships were consistent when the mean UPF intake was calculated instead of the median.

### 3.1. Ultra-Processed Food Intake by Population Subgroups

Lower consumption of UPF (% kcal and % g) was associated with being female, being from an ethnic minority background, being from a higher-income family, and being from southern England (Appendix A). Stratification of UPF content by income and meal type showed an income gradient, with lower-income children consuming the higher levels of UPF content (% kcal and % g) in both school meals and packed lunches. However, in primary schoolchildren who took a school meal, the income gradient in UPF content appeared to be less steep (Appendix A).

### 3.2. Composition of School Meals and Packed Lunches

In primary school, ultra-processed bread and snacks contributed to nearly half of the energy intake of packed lunches, compared to 13% kcal in school meals (Figure 1a). Conversely, a higher proportion of the energy of school meals consisted of minimally processed starchy foods as compared to packed lunches; however, a higher proportion of the energy of school meals was from UPF food groups, such as fast foods, puddings and desserts, and ready-to-eat foods, as compared to packed lunches. The distribution was similar in secondary schools, except that there was a lower proportion of energy from minimally processed fruit and vegetables and a higher proportion of energy from ultra-processed breads in school meals as compared to packed lunches in primary school.

When the contribution of food groups to the total weight of food consumed at lunch (% g) was explored (Figure 1b), the distribution of food groups was altered. This was due to the differing energy densities of food and drink products. In primary schools, the lower intake of UPF(% g) in school meals was driven by a lower intake of ultra-processed drinks, bread, and snacks, and a higher intake of minimally processed meat, starchy foods and fruit and vegetables, when compared against packed lunches. In terms of food groups consumed, these overall differences between packed lunches and school meals persisted in secondary schools. However, there was a higher proportion of ultra-processed drinks in school meals in secondary schools than in primary schools, which appears to largely account for the overall higher level of UPF. The composition of packed lunches was similar in primary and secondary schoolchildren, with high levels of UPF foods, such as drinks, bread, and snacks.

### 3.3. Quantile Regression of Meal Type on Ultra-Processed Food Content

In the quantile regression analysis, the association between school meal type and UPF intake was tested (Table 4). For primary schoolchildren in a fully adjusted analysis (Model 2), median UPF % kcal was 19.6 percentage points (pp) lower (95%CI: −22.3, −17.5) and median UPF % g was 24.8 pp (95% CI: −28.1, −22.3) lower in school meals compared to packed lunches. The difference in UPF content between meal types was smaller among secondary schoolchildren compared to primary schoolchildren. The fully adjusted models showed that school meals had 11.1 pp lower median UPF % kcal (95%CI: −15.99, −6.96) and 11.6 pp lower median UPF % g (95% CI: −21.03, −6.51) than packed lunches in secondary school.

The marginal effects from the fully adjusted regression model with an interaction between meal type and income are displayed in Figure 2.

Overall, the lowest income group had a higher UPF intake than the higher income groups, and there were similar income gradients in the UPF intake of school meals and packed lunches. However, in primary school the income gradient for UPF content (% g) differed by meal type (Figure 2A). It was observed that there was an income gradient for the UPF content (% g) of primary schoolchildren’s packed lunches, but there was no evidence of a gradient for school meals. For example, the lowest income group in primary schoolchildren had a significant 20 pp difference in the median UPF (% g) content of their packed lunches (74.9% g; CI: 67.7, 82.2), compared to the highest income group (54.3% g; CI: 47.3, 61.3). However, this was not observed in school meals for primary schoolchildren, where there was no significant difference in UPF content between income groups (also see Appendix A). 

### 3.4. Logistic Regression of Meal Type on Minimally- and Ultra-Processed Food Groups

In general, school meals were more likely to contain MPF groups and less likely to contain UPF groups when compared against packed lunches, regardless of school phase (Figure 3). However, there were a few exceptions. For example, the consumption of ultra-processed drinks by primary schoolchildren was substantially lower with school meals compared with packed lunches (AOR 0.1; 95%CI: 0.1, 0.2), but there was no evidence of a difference in secondary schoolchildren (AOR 0.8; 95%CI: 0.6, 1.1). We also found that school meals were more consistently more likely than packed lunches to contain ultra-processed puddings and desserts, vegetables, and fast food across school phases (pudding: AOR 2.5; 95% CI: 2.1, 3.1, UPF vegetables: AOR 7.1; 95% CI: 4.6, 10.8, fast food: AOR 8.2; 95% CI: 6.4, 10.4)

### 3.5. Sensitivity Analyses

The findings were robust to additional adjustments for energy, lunch portion, and BMI (Appendix A). Additionally, the analyses were repeated after excluding the 1580 participants whose meal type was defined using a question over general school meal preference in the survey and recorded at the time of eating in the food diary to explore whether differences in meal-type definition could explain the results (Appendix A). Differences in the UPF content of school meals and packed lunches were comparable to the original findings after the sample exclusions. Finally, the analyses were also found to be robust to a bias of energy misreporting, as results were comparable after unreliable energy reporters (*n* = 485, 15%) were excluded from the analyses (Appendix A).

## 4. Discussion

We found that the UPF content in school lunches was high in both primary (72.6% kcal, 43.7% g) and secondary schools (77.8% kcal, 52.5% g). School meals had a consistently lower UPF content than packed lunches, and their UPF content increased with advancing age. In primary school, UPF % kcal was 20 pp lower and UPF % g was 25 pp lower in school meals compared to packed lunches in fully adjusted models. In secondary schools, UPF % kcal was 11 pp lower and UPF % g was 12 pp lower in school meals compared to packed lunches. Finally, there was a socioeconomic gradient observed in UPF content of lunches, whereby lower income children consumed higher levels of UPF; however, this was not observed for the school meals of primary schoolchildren.

UPF consumption may be higher during school lunchtimes than at other times in the day. We saw that the lunchtime intake of UPF was higher (primary schoolchildren 68.8% kcal; secondary 71.6% kcal) compared to estimates of daily UPF intake in British children (65.4% kcal) [1] and adolescents (67.8% kcal) [33] based on the same NDNS data. These two previous studies assessed the association between patterns of eating context and intake of UPFs, and found that the school context was associated with lower UPF consumption in children but not in adolescents. However, the studies were not restricted to only school days. Our study is the first to estimate UPF intake during school lunchtimes, excluding other locations, indicating a high level of consumption at school lunchtimes in all school phases and with all meal types and income groups studied. As high UPF intake in childhood has been associated with greater adiposity and blood lipid levels [34,35,36], it is concerning that school lunches are a source of high UPF intake in children.

While both meal types had high UPF content, the levels were greatest in packed lunches. UPF contributed to 20 pp and 11 pp less of the energy content of school meals compared to packed lunches in UK primary and secondary schoolchildren, respectively. UPFs have been engineered to be preferable for time- and budget-restricted families who need cheap and quick food that is readily accepted by young children [37,38], so they are likely to be high in packed lunches. Packed lunches have long been recognised as being of poor nutritional quality on average [39], but previous research has not considered the growing availability of UPFs and their potential health concerns. We advance the current understanding and show that school meals were more likely to contain the minimally processed version of a food and packed lunches were more likely to contain the ultra-processed version by disaggregating food groups by the degree of industrial processing. Nevertheless, our study also highlights several food groups that may be provided in ultra-processed forms in school meals, such as puddings, some vegetables (e.g., baked beans and processed peas), and fast foods.

It is important to emphasise that while school meals had lower levels of UPF than packed lunches, they were still high. Furthermore, we present evidence of income inequality for UPF content in school meals, especially in secondary schoolchildren. It is unclear whether this is due to individual differences in food choices or differences in the food that was on offer. For example, secondary schoolchildren are given more choice for their school lunches, but are also typically given more ‘on-the-go’ foods [40,41], which are likely to be ultra-processed. Furthermore, low-income children may be more likely to choose UPF at school than higher-income children, as UPFs are more heavily marketed in more deprived neighbourhoods [42] and may be more present in their home environment [43]. However, it is also possible that the inequality is due to variation in food on offer; some children may have a more limited choice and higher exposure to UPF than other children.

The WHO has identified public procurement of food is an under-utilised tool to improve dietary intake [44], and this could be used to regulate levels of industrial processing in school food. In Brazil, a school feeding programme in public schools requires that 75% of the food purchased must be minimally processed and 30% must be supplied from local sources [45]. Evidence indicates that this policy is associated with a lower intake of UPF foods [46], better diet quality [47] and lower obesity [48]. However, in the UK there is currently no maximum level of UPF allowed in school meals [49,50]. All UK school food procurement must abide by mandatory standards, and all food served in schools must meet food-based standards, specific to each of the four nations in the UK [49,50]. There have been calls for the Government to update their procurement rules [51], but these currently fail to recognise UPF. Including maximum levels of UPF in these two standards would help to ensure that healthy, minimally processed food can be offered at an affordable price in all schools nationally.

### Strengths and Limitations

This is the first study to describe the UPF content of children’s diets at school. We used national representative data to produce estimates generalisable to the UK population. The dataset contained detailed dietary and sociodemographic data; therefore, we were able to explore both the effect modification of school phase and income, which has not been previously described in the school lunches of UK children. Additionally, the detailed dietary information in the NDNS dataset allowed for the NOVA categorisation to be applied to all food items, which was coded independently by two researchers.

There are a few limitations to note. The meal type of participants who did not record their meal type in the food diary (48%) was assumed by answers on school lunch preference in the survey. Sensitivity analysis determined that the study’s findings were robust to differences in measurement of meal type (Appendix A). Dietary diaries for children <12 years old were recorded by a proxy, which may introduce systematic measurement bias in the data of primary and secondary schoolchildren. Self-reported dietary data a often under-reported; therefore, the difference between primary and secondary schoolchildren may be underestimated. However, results were found to be robust to excluding energy intake misreporting. Furthermore, the outcome was assessed relative to total lunchtime energy and grams intake, and the analyses were additionally adjusted for energy intake, total grams, and BMI in sensitivity analysis to ensure there were no confounding effects from systematic differences in energy intake or in body size. The data pooled the years 2008–2017, limiting the ability to model changes overtime; however, survey year was adjusted in the analysis. Finally, as it was an observational study, there may be unobserved or residual confounding, which may introduce bias.

## 5. Conclusions

This study demonstrates that British schoolchildren consumed high levels of UPF in their school lunches. Higher intake of UPFs was more likely for children consuming packed lunches, children in secondary school, or children from low-income households. As such, this study highlights the need for a renewed focus and better guidance that considers the quality and level of industrial processing in food served in schools to ensure the dual benefit of encouraging school meal uptake and equitably improving children’s diet.

## Figures and Tables

**Figure 1 nutrients-14-02961-f001:**
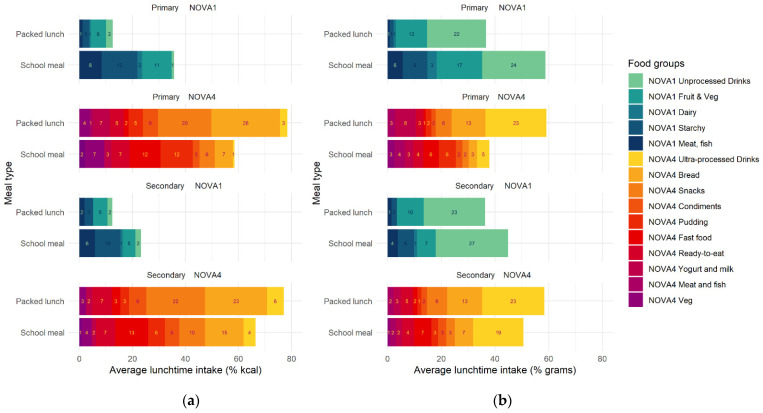
Average contribution of minimally and ultra-processed food groups to total food consumed at school lunch, stratified by meal type and school phase. (**a**) Contribution of minimally and ultra-processed food groups to total lunchtime energy (% kcal); (**b**) contribution of minimally and ultra-processed food groups to total lunchtime grams (% g). Note: NOVA1 = minimally processed; NOVA4 = ultra-processed foods; unprocessed drinks = water, juice etc; starchy = grains, rice and legumes; processed drinks = carbonated beverages; ultra-processed vegetables = baked beans and processed peas.

**Figure 2 nutrients-14-02961-f002:**
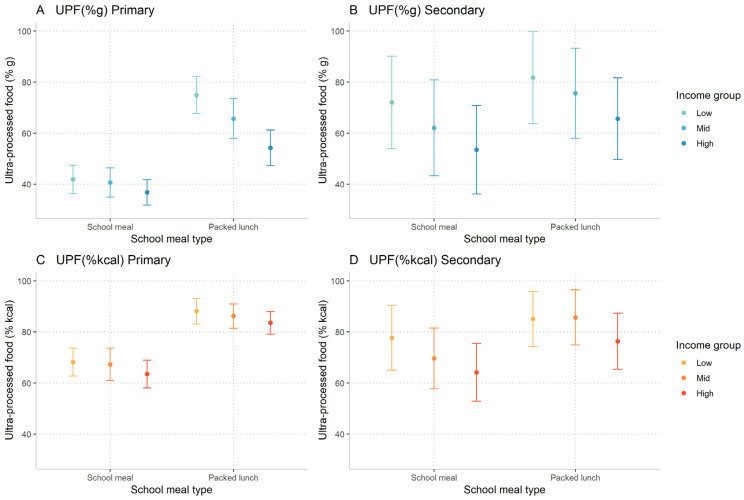
Marginal effects from a quantile regression of ultra-processed food intake at lunch with an interaction between meal type and income group, stratified by school phase. (**A**) UPF as % g in primary schoolchildren. Interaction between meal type and income: (Low # school meal—reference; Mid # school meal *p* = 0.08; High # school meal *p* = 0.01); (**B**) UPF as % g in secondary schoolchildren. Interaction between meal type and income: (Low # school meal— reference; Mid # school meal *p* = 0.68; High # school meal *p* = 0.77); (**C**) UPF as % kcal in primary school. Interaction between meal-type and income: (Low # school meal— reference; Mid # school meal *p* = 0.76; High # school meal *p* = 0.96); (**D**) 3D UPF as % kcal in secondary schoolchildren. Interaction between meal-type and income: (Low # school meal— reference; Mid # school meal *p* = 0.12; High # school meal *p* = 0.39). Note: Estimates are adjusted for age, sex, survey year, ethnicity, region, and IMD. Coefficients from interaction model given in Appendix A.

**Figure 3 nutrients-14-02961-f003:**
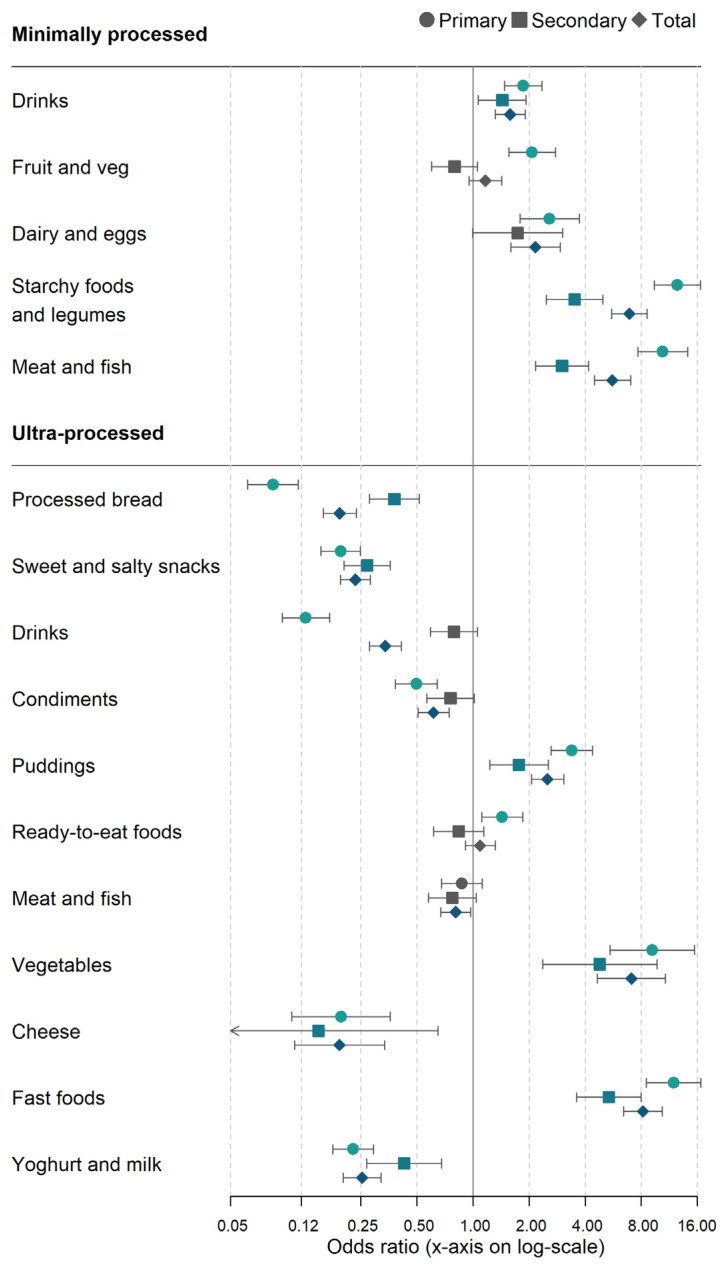
Logistic regression of the likelihood of consuming minimally and ultra-processed food groups by meal type (school meals vs. reference of packed lunches) and school phase. Note: Fully adjusted regression model, covariates listed in Appendix A. Unprocessed drinks = water, juice etc; processed drinks = carbonated beverages; ultra-processed vegetables = baked beans and processed peas.

**Table 1 nutrients-14-02961-t001:** Description of the four main groups within the NOVA food classification system.

Main Group	Description
NOVA 1 Unprocessed and minimally processed food (MPF)	Whole foods that have undergone no or minimal processing; e.g., fresh fruit and vegetables and their juices, cooked rice, plain milk or grilled fish.
NOVA 2 Processed culinary ingredients	MPF substances that are obtained directly from Group 1 foods (e.g., butter) or from nature (e.g., salt) and are used in food preparation
NOVA 3 Processed food	Foods that have undergone higher levels of processing and are manufactured with culinary ingredients; e.g., cheese, jam, or bread made from flour, water, and salt.
NOVA 4 Ultra-processed food (UPF)	Foods that are industrial formulations of substances derived from foods, and that contain cosmetic additives and little, if any, whole foods (e.g., carbonated beverages, French fries, or manufactured bread).

**Table 2 nutrients-14-02961-t002:** Description of the subsidiary groups within the NOVA food classification system.

Subsidary Group	Definition
**NOVA 1—minimally processed food (MPF)**	
Drinks	Water, coffee and tea, fresh fruit juices and smoothies
Fruit and vegetables	Fruit, vegetables, fungi, nuts and seeds
Dairy and eggs	Milk, plain yoghurt, eggs
Starchy foods and legumes	Grains, legumes, pasta, homemade pies and pastries
Meat and fish	Fish, poultry, red meat, pies and pastries with meat or fish, seafood
**NOVA 4—Ultra-processed (UPF)**	
Processed bread	Industrially manufactured bread
Sweet and salty snacks	Industrially manufactured cakes, pies, biscuits, sweet snacks, salty snacks (crisps)
Drinks	Soft drinks (high and low calorie) and fruit drinks
Condiments	Sauces, dressings, gravy, spread, margarine
Puddings and desserts	Ice cream, ice pops, desserts, sweet spreads and icing, artificial sugars and sweeteners
Ready-to-eat foods	Pasta and rice dishes (ready-to-eat/heat), egg and cheese dishes (ready-to-eat/heat), bacon/sausages dishes (ready-to-eat/heat), meat dishes (ready-to-eat/heat) Chicken/turkey dishes (ready-to-eat/heat), fish dishes (ready-to-eat/heat), vegetables dishes (ready-to-eat/heat), meat alternatives Potato dishes (ready-to-eat/heat), instant and canned soups, industrially manufactured meat pies and pastries
Meat and fish	Processed meat and fish (bacon, ham)
Vegetables	Processed vegetables (baked beans, processed peas)
Cheese	Processed cheese and cheese products
Fast foods	Pizza, French fries and other potato products, sandwiches and hamburgers
Yoghurt and milk	Industrially manufactured yoghurts and milk drinks

**Table 3 nutrients-14-02961-t003:** Sample characteristics of study participants from the National Diet and Nutrition Survey stratified by school phase and school meal type.

	Primary School Children (*n =* 1895, 57%)			Secondary School Children (*n =* 1408, 43%)			Total (*n =* 3303)		
Variable	School Meals	Packed Lunches	*p* ^1^	School Meals	Packed Lunches	*p* ^2^	School Meals	Packed Lunches	*p* ^3^
*n* (%)	928 (49.1)	967 (50.9)		654 (44.7)	754 (55.3)		1582 (47.1)	1721 (52.9)	
**Age, M (SD)**	7.2 (2.0)	7.7 (2.1)	<0.001 ^b^	13.7 (1.9)	14.2 (1.9)	0.001 ^b^	9.9 (3.8)	10.8 (3.8)	<0.001 ^b^
**Sex, *n* (%) ^4^**			0.03 ^a^			0.17 ^a^			0.55 ^a^
Male	480 (49.4)	525 (55.6)		327 (52.9)	340 (48.3)		807 (50.9)	865 (52.2)	
Female	448 (50.6)	442 (44.4)		327 (47.1)	414 (51.7)		775 (49.1)	856 (47.8)	
**Ethnicity, *n* (%)**			0.009 ^a^			0.02 ^a^			<0.001 ^a^
White	791 (79.6)	861 (85.6)		573 (78.9)	673 (85.5)		1364 (79.3)	1534 (85.5)	
Ethnic minorities	137 (20.4)	106 (14.4)		81 (21.1)	81 (14.5)		218 (20.7)	187 (14.5)	
**Household income, *n* (%)**				0.002 ^b^			0.10 ^b^		
Low	336 (37.5)	268 (31.0)		250 (43.1)	247 (36.4)		586 (39.9)	515 (33.5)	
Mid	271 (27.0)	370 (36.6)		199 (28.1)	261 (33.9)		470 (27.4)	631 (35.3)	
High	321 (35.5)	329 (32.4)		205 (28.9)	246 (29.6)		526 (32.7)	575 (31.1)	
**IMD, *n* (%)**			0.10 ^a^			0.12 ^a^			0.01 ^a^
1 (Least deprived)	188 (20.8)	229 (22.5)		143 (21.4)	178 (23.7)		331 (21.1)	407 (23.1)	
2	162 (15.5)	166 (19.1)		126 (20.9)	158 (21.8)		288 (17.8)	324 (20.4)	
3	199 (19.7)	207 (17.7)		127 (14.7)	156 (18.9)		326 (17.6)	363 (18.3)	
4	170 (19.8)	197 (21.9)		123 (22.1)	148 (20.8)		293 (20.8)	345 (21.4)	
5 (Most deprived)	209 (24.1)	168 (18.7)		135 (20.9)	114 (14.8)		344 (22.8)	282 (16.9)	
**Country, *n* (%)**			0.64 ^a^			0.29 ^a^			0.30 ^a^
England	555 (82.9)	543 (83.0)		366 (84.1)	460 (87.2)		921 (83.4)	1003 (85.0)	
Scotland	135 (8.8)	166 (9.2)		87 (6.3)	96 (5.5)		222 (7.7)	262 (7.5)	
Wales	120 (5.2)	118 (4.3)		88 (5.9)	89 (4.3)		208 (5.5)	207 (4.3)	
N. Ireland	118 (3.1)	140 (3.5)		113 (3.7)	109 (3.0)		231 (3.4)	249 (3.2)	
**UPF % kcal, med (Q25,Q75)**	61.0 (44, 75)	81.2 (71, 91)	<0.001 ^c^	70.1 (48, 89)	83.5 (65, 99)	0.001 ^c^	64.0 (45, 80)	82.1 (68, 94)	<0.001 ^c^
**UPF (% kcal), M (SD)**	58.7 (22.5)	78.4 (17.4)	<0.001 ^b^	66.0 (28.3)	76.2 (25.7)	0.001 ^b^	61.8 (25.4)	77.4 (21.7)	<0.001 ^b^
**UPF % g, med (Q25,Q75)**	35.4 (23, 49)	59.9 (36, 82)	<0.001 ^c^	45.8 (22, 80)	58.6 (29, 91)	0.01 ^c^	37.7 (23, 60)	59.7 (23, 60)	<0.001 ^c^
**UPF (% g), M (SD)**	38.2 (20.9)	59.8 (27.0)	<0.001 ^b^	50.3 (32.0)	57.8 (32.7)	0.001 ^b^	43.3 (26.8)	58.9 (29.8)	<0.001 ^b^

Note: Packed lunch—food brought from home; school meals—meals bought at the school canteen; M—mean; SD—standard deviation; IMD—Index of Multiple Deprivation. IMD is an area-based measure of deprivation; UPF—ultra-processed food; med—median; IQR—interquartile range. ^1^ Significance test between primary school meals and packed lunches. ^2^ Significance test between Secondary school meals and packed lunches. ^3^ Significance test between school meals and packed lunches in total sample. ^4^ Percentage of covariates within total packed lunch or school-meal users. ^a^ Chi-square test. ^b^
*t*-test. ^c^ Rank-sum test.

**Table 4 nutrients-14-02961-t004:** Quantile (median) regression exploring the association between school meal type and ultra-processed food intake, stratified by school phase.

	Primary Schoolchildren				Secondary Schoolchildren			
	Model 1 ^1^		Model 2 ^2^		Model 1 ^1^		Model 2 ^2^	
**Variable**	Coef. (95% CI)	*p*	Coef. (95% CI)	*p*	Coef. (95% CI)	*p*	Coef. (95% CI)	*p*
**UPF (% g)**								
Packed lunches	(ref)		(ref)		(ref)		(ref)	
School meals	−24.41 (−29.43, −21.45)	<0.001	−24.78 (−28.12, −22.3)	<0.001	−15.26 (−22.49, −7.21)	<0.001	−11.64 (−21.03, −6.51)	<0.001
**UPF (% kcal)**								
Packed lunches	(ref)		(ref)		(ref)		(ref)	
School meals	−20.42 (−22.72, −17.68)	<0.001	−19.64 (−22.26, −17.48)	<0.001	−13.07 (−16.49, −9.63)	<0.001	−11.05 (−15.99, −6.96)	<0.001

^1^ Minimally adjusted model—age and sex. ^2^ Fully adjusted model—age, sex, ethnicity, survey year, region, IMD, and income. (ref) = reference category.

## Data Availability

The dataset analysed in the study is available in the UK data service (SN: 6533). Accessed from http://doi.org/10.5255/UKDA-SN-6533-17 (accessed on 1 June 2022).

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
