# Peer review of "The Ultra-Processed Food Content of School Meals and Packed Lunches in the United Kingdom"

_nutrients, 2022, doi:10.3390/nu14142961_

Round 1

Reviewer 1 Report

I think this is a VERY nice study and very important.  Most of my comments address reporting issues that I think will make it easier for the reader to see the importance of your work.

You did not assess content you determined content.  Throughout I would suggest double checking the subject verb alignment, making sure that the subject can actually carry out the action.  I have provided examples below.

Line 20 – regression was used to determine the association.  Assess is the not the correct verb.

Does the UK have guidelines that restrict the amount of sodium, sugar, and fat that can be present in the overall meal? This needs to be included in the Introduction.  Also, are meals provided free of charge?

Line 57 – school food differs among children by household income.

Line 61 – provide a representative . . .

Line 63 – uses (not used) as it is a rolling survey

How are the dietary assessment tools administered? 

Line 71 – N=4800 because this is your total sample

When was the data collected?  Was all data collected during a specific time frame?  This is not clear.  Lines 78-81 illustrate the importance of sharing this as holidays and other factors could impact reporting.

I would suggest converting lines 96-106 be converted to a table.  This is a very nice classification system and it is more difficult to understand it if written as narrative vs. being presented as text.  The same applies to subgroups within each of the four categories.  I see that this is in the supplemental materials but I think it should be in the main text.  Also, review what tables/figures should be included in text vs. supplemental.  Only include those tables in text that support answering your research questions.

Lines 167-168 – what do you mean by took a school meal?  Do you mean once, multiple times per week?

Table 1 – what do you mean by least deprived and most deprived? What is UPF?  Table was stand alone.

Line 191 – ethnic minorities are more likely to consume lesser UPF.  This is VERY interesting.  Is it because they are more likely to bring a lunch because the foods that are served are not culturally appropriate?  Also, a subset of ethnic minorities might be in greatest need for getting meals through school so focusing on offering culturally appropriate foods could be a next step.

Line 203 – puddings is a very British term : ) this journal has international scope so might need to explain.

Lines 201- 203 – School meals contained a higher proportion of energy as . . . but also higher proportion of kcals attributed to fast foods, puddings, . . .

School meals to not have an intake . . . school meals provide kcals so the verbs need to match the subject.

Lines 207-208 – this is not clear – do you mean weight of student or weight of food?

Again, the Figure 1 will make more sense if you include a table summarizing the NOVA categories

What were your research questions/hypotheses?

Line 286 – a food diary is not a 3-day recall.  A food diary is a prospective form of data collection whereas a recall is retrospective.  This is not clear.

Line 310 – estimate UPF intake during school lunch time.

Line 316 – what do you mean by 20 pp and 11pp?

When you refer to school meals I assume that you mean the meals provided by the school?  Are these provided free of charge?  This needs to be clear as the journal targets an international audience.

Line 357 – data cannot explore, data can be used to explore

Another limitation is that the time during which data was collected was not consistent.  For example, if the data was collected when benefit checks arrived meals could be higher quality than later in the month when resources run out.

The only tables/figures that should be included are those that support answering your research questions/hypotheses. 

Author Response

Please see response in letter attached

Reviewer 2 Report

It is a very interested project and important for childen's health while they are growing up.

Generally, students will have same school meals provided in UK, how can you obtain student's meal difference in one school based on their family's household income?  If you have sound evidence for this, I think it will be even better.

Author Response

Please see response in letter attached
